# Medication Adherence to Psychotropic Medication and Relationship with Psychiatric Symptoms among Syrian Refugees in Turkey: A Pilot Study

**Gül Dikeç** [1],* and **Kübra Timarcıoğlu** [2]

1   Department of Nursing, Faculty of Health Sciences, Fenerbahce University, 34758 Istanbul, Turkey
2   Turkish Red Crescent Community Health Centre, 63300 Sanlıurfa, Turkey
*   Correspondence: gul.dikec@fbu.edu.tr; Tel.: +90-216-910-19-07

**Abstract:** Background: Due to experiencing traumatic and stressful events, refugees are at risk of having mental disorders. The refugees might need to use psychotropic agents to treat mental disorders. It is essential to understand this population's adherence rate to psychotropic medication. Aim: This study aimed to evaluate adherence to psychotropic medication and the relationship with psychiatric symptoms among Syrian refugees in Turkey. Method: The study design was descriptive and correlational. The study was conducted with 55 Syrian refugees attending a community health center in southern Turkey. The data was collected with General Medication Adherence Scale–Arabic Version (GMAS–AR) and Brief Psychiatric Inventory (BPI). Results: 78.2% ($n$: 43) of the participants' medication adherence was poor, and a significant negative and weak correlation was found at 0.01 level between GMAS–AR and BPI (r: −0.37). According to multiple linear regression analysis, the model with gender and BPI was significant, and this model explained 19% of GMAS–AR total points ($p < 0.001$). Conclusion: Mental health professionals should test the effectiveness of psychosocial interventions that increase adherence to treatment for the cultural characteristics of societies.

**Keywords:** adherence; mental disorders; psychiatric nursing; refugee; trauma

## 1. Introduction

In recent years, people have been leaving their countries due to the world's increasing violence, war, disaster, and human rights violations. The countries they move to try to establish a new order, provide social support and help refugees cope with their traumatic experiences [1,2]. According to the International Organization of Migration (IOM) and the United Nations report, Turkey is one of the countries that receive the most refugees [1,3–5]. Turkey, which has opened its borders to individuals from Afghanistan, Pakistan, and Iran, probably has one of the highest rates of transitory migrants and has the highest Syrian refugee population [6].

Undoubtedly, refugees are the most in need of assistance among the entire migrant population. During the war, refugees often experience the loss of family members, sexual and physical abuse, and lack of shelter and nutrition. They also experience many social difficulties, such as language barriers, stigmatization, social isolation, economic problems, and adjustment difficulties after displacement. Therefore, these individuals are at higher risk for developing mental disorders [1,2,4,5,7–9]. It has been determined that they are at risk regarding anxiety disorders, depressive disorders, and psychotic disorders [1,2,10,11]. Due to these emerging mental disorders, they need mental health services more frequently [12], while they receive less psychotherapy [7]. Health services, especially mental health services, are provided through government services in Turkey and surrounding countries, alongside humanitarian programs and the projects of non-government organizations for refugees [4,13]. Although there is no study with a large sample on the use of mental health services by Syrians in Turkey, it is noteworthy that this rate was low in the studies. In

Fuhr et al. [4], a study conducted in Istanbul showed that while 249 participants (15% of the total) screened positive for either symptom of post-traumatic stress disorders, depression, and anxiety, only 22 (9%) of them sought mental health care and 219 participants did not (88%). The treatment gap was found for anxiety (90%) and depression (88%), too.

Treatment adherence could relate to patients' compliance with health-related recommendations [14]. However, patients sometimes do not follow their diet or treatment program as recommended. Non-adherence to treatment for mental disorders is a frequently encountered problem [15]. Many factors affect medication adherence. These are patient-related (psychotic symptoms, previous non-adherence experience, inadequate social support, age, education, etc.) [14,16], treatment-related (complexity of treatment regimen, side effects, etc.) [14,17], health care professionals-related (lack of cooperation with the treatment team, lack of therapeutic relationship, etc.), and system-related (short time allocated to patients, bureaucratic obstacles, etc.) [17]. Non-compliance with treatment causes exacerbation of psychiatric symptoms and relapses [16], affecting hospital readmission. Re-hospitalization increases patient care costs [14]. For this reason, it is vital to evaluate treatment adherence and to improve it with appropriate psychosocial interventions [4].

The ethnicity of minority groups is one of the influencing factors that may negatively impact medication adherence [8,14]. Although mental disorders are common in refugees, it is reported that their adherence to psychotropic agents used for the treatment of mental disorders is poor [7]. In a qualitative study conducted in Turkey, Doğan [12] found that Syrian individuals had difficulties getting an appointment, expressing themselves, and obtaining medications while receiving mental health services. It was determined that they could not get the psychotropics from pharmacies, did not take them when they could not find the medications, or tried to use them by reducing the dose. At this point, it can be said that the adaptation of refugees toward mental health treatment is affected by external factors.

On the other hand, culture is the most important factor shaping the perception of illness and the perspective toward treatment. Culture may cause treatment adherence problems by affecting the individual's internal factors toward treatment [2]. Sacha [2] found that the predictors of non-adherence to treatment were not examined in the studies reviewed; however, the language issues of refugees indicated that barriers to receiving health services might be related to this situation. For this reason, there is a need to examine the factors that affect refugees' treatment adherence and specific transcultural intervention programs for this population [8].

Since Turkey is the leading refugee-hosting country, several studies have determined mental health problems and comorbidities among Syrian refugees [9]. However, no research on medication adherence to psychotropic medication and effecting factors of medication adherence to psychotropic agents among refugees, especially Syrian refugees, has been found in the literature. Determining adherence to psychotropic treatment and related factors in Syrian refugees is crucial in developing interventions that can enhance treatment adherence. Therefore, this study aimed to determine the relationship between medication adherence to psychotropic agents and psychiatric symptoms among Syrian refugees who attended a community health center in Turkey.

*Research Questions*

The research sought answers to the following two fundamental questions:

1. What is the level of psychotropic medication adherence among Syrian refugees?
2. Is there a relationship between the psychiatric symptoms of Syrian refugees and their treatment adherence?

## 2. Materials and Methods

### 2.1. Study Design

This study was planned as a descriptive and correlational study design. It was a pilot study prior to psychosocial interventions to increase the medication adherence of Syrian refugees in Turkey.

### 2.2. Settings

This study was carried out in the community health center of a non-governmental organization located in southern Turkey between 2 August 2022 and 26 September 2022.

### 2.3. Participants

Syrian refugees who attended the community health center were included in the study. Inclusion criteria were being literate in Arabic, agreeing to participate in the study, being between the ages of 18–65, being diagnosed with any mental disorder according to DSM-V, and using any oral or parenteral psychotropic medications for the last three months. Exclusion criteria were being of non-Syrian origin, illiteracy in Arabic and refusal to participate in the study after being informed, not having any psychiatric disorder diagnosis according to DSM-V, only seeking counseling or applying to the institution due to physical problems, and using psychotropic medication less than the last three months. In the study, the sample calculation of the known population was made. The population of the study consisted of 74 individuals aged 18 and over, of Syrian nationality, and using psychotropic drugs in the designated center. Sample 62 was determined with a 5% margin of error and a 95% confidence interval in the sample calculation. Fifty-five people agreed to participate in the study within the specified time. According to the sampling calculation, 88.70% of the sample was reached.

### 2.4. Measurements

The Personal Information Form, which includes sociodemographic and medication-related information, the General Medication Adherence Scale–Arabic Version, and the Turkish Version of the Brief Symptom Inventory, were used to collect the data.

#### 2.4.1. Personal Information Form

For the study, the form was prepared by the researchers by scanning the literature [2,17,18] and includes the sociodemographic, mental disorders, and treatment-related characteristics of refugees. This form has a total of 11 questions.

#### 2.4.2. General Medication Adherence Scale–Arabic Version (GMAS–AR)

Naqvi [18] developed this scale to measure medication adherence in chronic conditions, and the scale was adapted to different languages in Pakistan and Saudi Arabia [19] apart from English. It is a self-reporting scale with 11 items and a 4-point Likert scale. The total score is obtained by summing all the items in the scale and varies between 0–33 points. Those with a score of 27 and above have "good" treatment adherence; those who score 26 points or less are evaluated as "not good" for treatment adherence. There are no subscales in the scale. The Cronbach alpha coefficient of the Arabic version of the scale was 0.86. In this study, Cronbach's alpha coefficient of the scale was 0.73.

#### 2.4.3. Brief Symptom Inventory (BSI)

The Brief Symptom Inventory is a frequently used scale to evaluate mental health. The scale developed by Derogatis [20] consists of 53 items and 9 subscales. The subscales of BSI are somatization, obsessive-compulsive disorder, interpersonal sensitivity, depression, anxiety, hostility, phobic anxiety, paranoid thoughts, and psychoticism. The scale is a 5-point Likert scale and is a self-reporting scale. Items were graded as "0", "not at all," "4", or "too much." It is a widely used and practical scale translated into 26 languages, applied to the healthy population and individuals with mental disorders. The total score range of

the scale varies between 0–212, and an assessment is made that the higher the score, the higher the psychological symptoms. Subscale scores are obtained by dividing the sum of the scores given to the items forming each subscale by the number of items. Subscale scores can range from 0 to 4. The original scale reported that the subscale of Cronbach's alpha coefficients ranged between 0.71 and 0.85 [20]. Şahin [21] had done Turkish validity and reliability of the scale, and Cronbach's alpha coefficient for the total scale score was 0.96 and 0.55–0.86 for the subscales. In this study, Cronbach's alpha coefficient of the scale was 0.95. The individuals filled out the Brief Symptom Inventory through an interpreter.

*2.5. Ethical Considerations*

Ethics committee approval was obtained dated 9 March 2022 and numbered 09.2022 fbu from Fenerbahçe University Non-Interventional Clinical Research Ethics Committee before starting the study; written permission was obtained from the Turkish Red Crescent Academy, where the study was conducted. In addition, verbal and written consent was obtained from the individuals participating in the study and their legal representatives.

*2.6. Statistical Analysis*

The data were analyzed using the SPSS (Statistical Package for Social Sciences) for Windows 20.0 program. Descriptive numerical data of the study were analyzed using mean and standard deviation, minimum and maximum, and categorical data as numbers and percentages (%). Normality analysis was performed by examining Kurtosis and Skewness. Kurtosis and Skewness values were evaluated between $-1.96$ and $+1.96$. Due to the normal distribution of the data, the relationship between the scales was analyzed by Pearson correlation analysis, and the Mann-Whitney U test or Kruskal Wallis was used to compare scales and sociodemographic characteristics since the sample size in the eyes was <30. The relationship between the non-normally distributed scale variables and the scales was analyzed by Spearman Correlation analysis. Multiple linear regression was used to analyze the predictors of treatment adherence as scales were disturbed normally. All values were evaluated at $p < 0.05$ significance level.

**3. Results**

*3.1. Main Characteristics of Total Sample*

The sociodemographic characteristics of the individuals participating in the study are shown in Table 1. The results determined that 52.7% were female, 65.5% perceived their economic level as medium, 78.2% were married, and 81.8% were secondary or high school graduates. It was determined that 58.2% of the refugees worked in a job with income, and 89.1% had a child. When the characteristics of mental disorders were examined, it was determined that 45.5% were followed up with the diagnosis of anxiety disorder, 96.4% had not received any inpatient treatment before, and 63.6% used antidepressants. When GMAS–AR, and BSI total score averages were examined according to their sociodemographic characteristics, a significant difference was found between the groups according to gender and hospitalization ($p < 0.05$).

*3.2. Correlation between GMAS–AR and BSI*

GMAS–AR and BSI scale scores are given in Table 2. It was determined that 78.2% (n: 43) of the refugees had poor adherence to treatment. According to the Spearman Correlation analysis performed between the scales, a negative and weak statistically significant relationship was found.

*3.3. Predictors of GMAS–AR*

Multiple regression analysis was performed, and the Backward method was used to determine the effect of gender, hospitalization, and BSI on GMAS–AR. Accordingly, the established model was found to be statistically significant, and 22% of the GMAS–AR

total scored average. It was determined that the model established with gender and BSI explained 19%, and BSI explained 14% alone of the GMAS–AR total score average (Table 3).

**Table 1.** Sociodemographic Characteristics of Participants and Comparing GMAS–AR and BSI Total Mean Scores According to These Characteristics.

| Characteristics | Min–Max | Mean (SD) | GMAS–AR Test/*p* | BSI Test/*p* |
|---|---|---|---|---|
| Age | 19-59 | 36.60 (9.58) | r: 0.02 | r: −0.24 |
| Gender | n (%) | | | |
| Female [1] | 29 (52.7) | | Z: −2.49 | Z: −1.17 |
| Male [2] | 26 (47.3) | | *p* <0.001 2>1 | *p*: 0.24 |
| Economical Level | n (%) | | | |
| Moderate [1] | 26 (65.5) | | Z: −1.60 | Z: −2.10 |
| Poor [2] | 19 (34.5) | | *p*: 0.10 | *p*: 0.03 2>1 |
| Marital Status | | | | |
| Single | 5 (9.1) | | X²: 3.26 | X²: 3.10 |
| Married | 43 (78.2) | | | |
| Other | 7 (12.7) | | *p*: 0.19 | *p*: 0.21 |
| Education Level | | | | |
| Primary School [1] | 7 (12.7) | | X²: 6.94 | X²: 0.14 |
| Middle or High School [2] | 45 (81.8) | | *p*: 0.03 | |
| University [3] | 3 (5.5) | | 2>3 | *p*: 0.9 |
| Occupation | | | | |
| Yes | 23 (41.8) | | Z: −0.89 | Z: −1.29 |
| No | 32 (58.2) | | *p*: 0.36 | *p*: 0.19 |
| Having a Child | | | | |
| Yes | 49 (89.1) | | Z: −0.48 | Z: −0.50 |
| No | 6 (10.9) | | *p*: 0.62 | *p*: 0.61 |
| Diagnosis of Mental Disorders | | | | |
| Mood Disorders | 15 (20) | | X²: 4.28 | X²: 5.91 |
| Schizophrenia and Other Psychotic Disorders | 6 (10.9) | | | |
| Anxiety Disorders | 25 (45.5) | | *p*: 0.23 | *p*: 0.11 |
| Trauma-Related Disorders | 9 (16.4) | | | |
| | Min–Max | Mean (SD) | | |
| Duration of Living in Turkey (month) | 36–216 | 103.98 (28.90) | r: −0.54 | r: 0.50 |
| Duration of Having Mental Disorders (month) | 3–360 | 63.16 (62.33) | r: −0.15 | r: 0.19 |
| Duration of Using Psychotropic Medications (month) | 1–108 | 13.67 (20.94) | r: 0.09 | r: 0.24 |
| Hospitalization | n (%) | | Z: 3.93 | Z: 3.07 |
| Yes | 2 (3.6) | | | |
| No | 53 (96.4) | | *p*: 0.04 | *p*: 0.07 |
| Psychotropic Medications | | | | |
| Antipsychotics | 4 (7.3) | | X²: 0.09 | X²: 4.13 |
| Antidepressants | 35 (63.6) | | | |
| Antipsychotics + Antidepressants | 16 (29.1) | | *p*: 0.95 | *p*: 0.12 |

Min: Minimum, Max: Maximum, X²: Kruskal Wallis Test. r: Spearman Correlation, Z: Mann-Whitney U test.

**Table 2.** Total and Subscales Scores of GMAS–AR and BSI.

| Scales | Min–Max | Mean (SD) | Skewness | Kurtosis | r |
|---|---|---|---|---|---|
| GMAS–AR | 5–130 | 22.14 (5.12) | −0.59 | 1.02 | −0.37 ** |
| BSI | 8–174 | 103.80 (43.49) | −0.45 | −0.76 | |

**: Correlation is significant at the 0.01 level.

**Table 3.** The effects of sociodemographic variables and BSI on GMAS–AR.

| Dependent Variable | Independent Variables | B | ß | t | *p* | F | Model (p) | R² |
|---|---|---|---|---|---|---|---|---|
| GMAS–AR | Constant | 32.19 | | 4.69 | <0.001 | 5.02 | 0.004 | 0.22 |
| | BSI | −0.03 | −0.30 | −2.37 | 0.02 | | | |
| | Gender | 2.42 | 0.23 | 1.92 | 0.06 | | | |
| | Hospitalization | −5.05 | −0.18 | −1.46 | 0.14 | | | |
| GMAS–AR | Constant | 22.87 | | 8.71 | <0.001 | 6.30 | 0.004 | 0.19 |
| | BSI | −0.04 | −0.34 | −2.78 | 0.007 | | | |
| | Gender | 2.39 | 0.23 | 1.88 | 0.06 | | | |
| GMAS–AR | Constant | 26.72 | | 15.87 | <0.001 | 8.66 | 0.005 | 0.14 |
| | BSI | −0.04 | −0.34 | −2.94 | 0.005 | | | |

## 4. Discussion

In this study, which was conducted to examine the relationship between treatment adherence to psychotropics and psychiatric symptoms in a group of Syrian refugees in Turkey, most refugees who participated in the study had low treatment adherence. When the predictors of treatment adherence were examined, it was determined that psychiatric symptoms explained 14% of the total GMAS–AR score. It is stated in the literature that non-adherence to psychotropic medication increases the symptoms of mental disorders, causes relapses and recurrent hospitalizations, homelessness, lower quality of life, and increases morbidity and mortality [15]. Since using psychotropic medication helps individuals to manage and control symptoms, maintenance treatment is essential to prevent psychiatric symptoms and relapses [22]. Non-adherence is a challenge of mental disorders and is associated with adverse patient outcomes, including exacerbating psychiatric symptoms and impaired functionality [23]. Since non-adherence increases psychiatric symptoms, patients who have non-adherence could experience hospitalizations and emergency use [24]. Therefore, medication adherence to psychotropic medication has implications for prognosis positively and protects long-term functionality. At this point, it can be said that the study results were similar to the literature.

In this study, it was found that the adherence scores of men were higher than women's. According to Tan [14], non-adherence to psychotropic treatment is commonly observed in young people and men. Individual sociodemographic variable, such as gender, is still not fully understood regarding refugees' adherence [2], and different results were found in studies on gender and treatment adherence. In a meta-analysis, Semahegn [17] emphasizes an inconsistency between studies' results related to gender and treatment adherence [17]. It may be a cultural difference among Syrians. In addition, there were no differences according to the sociodemographic characteristics of participants. It might be related to the small sample size. Therefore, in future studies, it may be recommended to determine the predictors of adherence to treatment with a large sample, especially in Syrian refugees.

Refugees, who are more at risk in terms of mental disorders due to the traumatic events they have experienced, may need mental health and psychosocial support (MHPSS) more frequently. However, they may experience difficulties accessing MHPSS due to language barriers, such as a lack of translators, not knowing the language spoken in the country, and economic or structural problems [10]. Again, the language barrier, insufficient cooperation with the treatment team, and inability to inform about the treatment may also affect adherence to the treatment. In Turkey, Doğan [12] reports that Syrian refugees have difficulties obtaining medicine and are unable to receive services due to economic problems or inability to meet transportation expenses, which affects their adherence to treatment. Even though mental health services and medication treatments are entirely free in the center where the study was conducted, it is noteworthy that treatment adherence is low.

In this study, many factors may have influenced the low adherence rate of refugees to psychotropics. First, non-adherence to treatment may be associated with culturally negative attitudes toward medication and mental disorders in Syrian refugees [10,25]. Bradl [7] has

emphasized that the attitude toward medications is more negative among refugees from the Middle East than among refugees from Turkey and Central Europe in their study with refugees in Berlin. In the study of Fuhr [4], conducted with Syrian refugees in Istanbul, Turkey, they investigated the reasons for those with psychiatric symptoms not initiating the treatment they need. The participants did not believe that the treatment would reduce the symptoms, would be a waste of time and cost, or it would cause more expenditure. They did not know where and how to get help, and they thought that if they received treatment, society would label them and exclude them. The fact that mental disorders are still taboo and stigmatized in many countries and cultures negatively affects adherence to treatment [1,2]; their study of Syrian refugees living in Germany determined that self-stigmatization negatively affects the mental health outcomes of refugees. In this study, individuals may have accessed mental health services due to the trauma and stressors they experienced. Although they have started treatment, they might not have been able to continue due to stigmatization. It could be recommended that the investigation between medication adherence, attitude, and stigmatization in this population in future studies.

Secondly, refugees sometimes express their health problems first to their family members or relatives, and they may resort to religious treatment methods [25]. They may expect the clergy, who are described as healers and who have superpowers, to heal themselves. Fuhr [4] found that Syrian refugees determined that they shared their mental problems with their friends and families first and then told them to religious leaders in their study. Seeking a remedy other than medicine, a cultural feature, may also affect treatment adherence. Finally, most refugees in this study use antidepressants for anxiety or mood disorders. It has been thought that the effects of antidepressants do not start immediately [26], side effects occur first, and individuals think that the medications do not improve them.

## 5. Limitations

The results of this study were limited to the data collected on a group of Syrian refugees and cannot be generalized. Another limitation of the study, since it was conducted in a single center and was a pilot study, is its small sample size. In the study, since the GMAS–AR was in Arabic, it was ensured that the individuals filled out forms based on their declarations, but the BSI was filled out through an interpreter. In turn, this can affect the results of individuals. In future studies, it may be recommended to use standardized measurement tools that have been validated and reliable in the refugees' mother tongue.

## 6. Conclusions

This study was a pilot study to assess the treatment adherence of a group of Syrian refugees in Turkey and the relationship between adherence and psychiatric symptoms. The treatment adherence to the psychotropic medication of the Syrian refugees who participated in this study was low. There was a statistically significant negative correlation between the refugees' adherence to psychotropic treatment and their psychiatric symptoms. Ensuring that refugees in Turkey, whose number is increasing daily, benefit from mental health services and reduce their psychiatric symptoms is crucial in protecting individuals' functionality and reducing disability. At this point, increasing the treatment adherence of refugees using psychotropic medications is essential. Mental health professionals should develop or modify psychosocial interventions such as psychoeducational programs, telenursing interventions, adherence therapy (AT), cognitive behavioral techniques, and motivational interviewing [27] to increase adherence to treatment for the cultural characteristics of societies. The effectiveness of the interventions should be tested. The authors have planned to test the effectiveness of AT, which was a combined intervention on cognitive behavioral techniques, motivational interviewing, and psychoeducation. AT was a collaborative and patient-centered program developed by Gray [22]. Six elements form the therapy: assessment; medication problem-solving; a medication timeline; exploring ambivalence; discussing beliefs and concerns about medication; and using medication in the future. In a systematic review and meta-analysis in which six studies were evaluated, it

was determined that AT reduces psychiatric symptoms [27], stating that AT could be used to increase adherence to treatment in their guideline. As AT is patient-center and has a flexible structure, in the next step, the authors should adapt AT to the Syrian culture. When the culturally sensitive skills of mental health professionals are developed, a therapeutic alliance can be achieved, and the causes of non-adherence to treatment can be explored in depth.

**Author Contributions:** Conceptualization, G.D. and K.T.; methodology, G.D.; formal analysis, G.D.; data collection and curation, K.T.; writing—original draft preparation, G.D.; writing—review and editing, G.D. and K.T. All authors have read and agreed to the published version of the manuscript.

**Funding:** This research received no external funding.

**Institutional Review Board Statement:** The study was conducted according to the guidelines of the Declaration of Helsinki and approved by the Fenerbahce University Non-Interventional Clinical Research Ethics Committee of the University of Health Sciences (dated 9 March 2022 and numbered 09.2022fbu). An institutional permit was obtained from the Turkish Red Crescent Academy.

**Informed Consent Statement:** Written consent was obtained from the individuals participating in the study and their legal representatives.

**Data Availability Statement:** The data presented in this study are available on request from the corresponding author.

**Acknowledgments:** The authors would like to thank all participants. Also, the authors have attributed this study to all the people who lost their lives in the two major earthquakes that occurred in Kahramanmaraş on 6 February 2023, to the Timarcıoğlu family and to the traumatized individuals.

**Conflicts of Interest:** The authors declare no conflict of interest.

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
