# Peer review of "Medication Adherence to Psychotropic Medication and Relationship with Psychiatric Symptoms among Syrian Refugees in Turkey: A Pilot Study"

_traumacare, doi:10.3390/traumacare3010005_

Round 1

Reviewer 1 Report

Thank you for your invitation reviewing the manuscript titled “Medication Adherence to Psychotropic Medication and Relationship with Psychiatric Symptoms among Syrian Refugees in Turkey: A Pilot Study”.

Please refer to the comments below.

 Abstract

 Please add the conclusion in the abstract.

 Introduction

 The background should include the population and mental health services utilization rate of Syrian Refugees to substantiate the need for the research problem.

 Line 77  Although Turkey is the leading refugee-hosting country, no study or rate on medication adherence to psychotropic medication and the relationship between adherence among refugees, especially Syrian refugees, has been found in the literature.

The meaning of this sentence is unclear about what variables were being tested for their relationship among refugees.

Line 80  This study aimed to examine the relationship between medication adherence to psychotropic agents and psychiatric symptoms in Syrian refugees who  were followed in a community health center in Turkey.

Materials and Methods

 Line 163  Please state how to handle the non-normality of data distribution for statistical tests with Multiple linear regression.

 How can it explain that gender was the predictor while the sig. level of gender is 0.06 in the regression analysis?

 The steps and each model of multiple linear regression should be elaborated.

 Results and discussions

 The sample size is too small to conclude. If it is a pilot study, what are the next steps? The authors should revisit the purpose of doing this pilot study and tie up with a conclusion that what recommendations will be made for the actual study.

 The authors should explain why there is a negative relationship between mental symptoms and adherence.

 The authors should please explain which gender predicts adherence and why.

The authors should also further explain why and how mental symptoms predict adherence in this study. In addition, explain the reasons why other demographic factors did not relate to adherence with reference to the characteristics of the participants.

Author Response

Reviewer I Comments

Response to Reviewer I

Thank you for your invitation reviewing the manuscript titled “Medication Adherence to Psychotropic Medication and Relationship with Psychiatric Symptoms among Syrian Refugees in Turkey: A Pilot Study”. Please refer to the comments below. 

Thank for your contributions.

 Abstract - Please add the conclusion in the abstract.

The conclusion sentence has been added to the end of the abstract.

Introduction- The background should include the population and mental health services utilization rate of Syrian Refugees to substantiate the need for the research problem.

Since we could not find any rate of mental health usage of Syrian Refugees, we shared one of the study results in lines 49-55 about this problem.

Line 77 Although Turkey is the leading refugee-hosting country, no study or rate on medication adherence to psychotropic medication and the relationship between adherence among refugees, especially Syrian refugees, has been found in the literature.

The meaning of this sentence is unclear about what variables were being tested for their relationship among refugees. 

We tried to clarify this sentence in lines 86-92.

Since Turkey is the leading refugee-hosting country, several studies have determined mental problems and comorbidities among Syrian refugees [9]. However, no research or rate on medication adherence to psychotropic medication and effecting factors of medication adherence to psychotropic agents among refugees, especially Syrian refugees, has been found in the literature.

Line 80 This study aimed to examine the relationship between medication adherence to psychotropic agents and psychiatric symptoms in Syrian refugees who were followed in a community health center in Turkey.

We tried to clarify this sentence in lines 90-92.

Therefore, this study aimed to determine the relationship between medication adherence to psychotropic agents and psychiatric symptoms among Syrian refugees who followed in a community health center in Turkey.

Materials and Methods 

Line 163 Please state how to handle the non-normality of data distribution for statistical tests with Multiple linear regression. 

We added the sentence "Normality analysis was performed by examining Kurtosis and Skewness.” in the statistical analysis section and gave the Kurtosis and Skewness value in Table 2. (lines 177-179, 185)

How can it explain that gender was the predictor while the sig. level of gender is 0.06 in the regression analysis?

There was a mistake. Thank for your attention.

The steps and each model of multiple linear regression should be elaborated. 

Each model was explained in the line 217 and Table 3.

Results and discussions 

The sample size is too small to conclude. If it is a pilot study, what are the next steps? The authors should revisit the purpose of doing this pilot study and tie up with a conclusion that what recommendations will be made for the actual study. 

We give more details about the project's next step in the conclusion section (lines 312-325).

The authors should explain why there is a negative relationship between mental symptoms and adherence. 

We explained the relationship between psychiatric symptoms and non-adherence. (lines 229-232)

The authors should please explain which gender predicts adherence and why. 

Since the p-value>0.05 was not a predictor of adherence, there was a significant difference between the gender. We discussed this point in the lines 239-247.

The authors should also further explain why and how mental symptoms predict adherence in this study. In addition, explain the reasons why other demographic factors did not relate to adherence with reference to the characteristics of the participants.

We explained the relationship between psychiatric symptoms and non-adherence. (lines 229-232)

We discussed that other demographic factors did not relate to adherence, it might be related to small sample size in the discussion section (lines 247-247).

Reviewer 2 Report

This is a study investigating the association between mental health and medication adherence in Syrian Refugees in Turkey. The paper is well written and of interest for the journal. However, several changes are recommended before considering it for publication.

Abstract.

1- I recommend to organize the abstract into the common structure: background, aims, methods, results and conclusions. The introduction section of the abstract is too long. It should be more concise.

2- The last part of the abstract is mainly a result. It lacks a conclusion sentence.

Introduction

1- In lines 47, medication adherence is mainly introduced. I recommend to explain main factors influencing mental health; for instance, social determinants of mental health, before introducing the medication adherence as a potential factor.

2- The main aims or objectives of the study should be detailed in a separated section.

Materials and methods.

1- All subsections of methods should be numbered. Study design and participants can be described in a unique section.

3- Data analysis should be better renamed as "Statistical analysis".

Results

1- The results section can be divided into several subsections. The first part, can be renamed as "Main characteristics of the total sample", with reference to Table 1.

Conclusions

1- Treatment adherence is low. I recommend to draw some recommendations about potential interventions.

Author Response

Reviewer II Comments

Response to Reviewer II

This is a study investigating the association between mental health and medication adherence in Syrian Refugees in Turkey. The paper is well written and of interest for the journal. However, several changes are recommended before considering it for publication.

Thank you for your contributions. The manuscript has been enhanced with your recommendations.

 Abstract 

 1- I recommend to organize the abstract into the common structure: background, aims, methods, results, and conclusions. The introduction section of the abstract is too long. It should be more concise.

2- The last part of the abstract is mainly a result. It lacks a conclusion sentence.

We have reorganized the abstract into the background, aim, method, result, and conclusion parts. The introduction part of the abstract was truncated, and the conclusion sentence was added.

Introduction

1- In lines 47, medication adherence is mainly introduced. I recommend to explain main factors influencing mental health; for instance, social determinants of mental health, before introducing medication adherence as a potential factor.

The introduction section added potential social factors affecting the refugee's mental health. (lines 38-43).

2- The main aims or objectives of the study should be detailed in a separated section.

Line 80 This study aimed to examine the relationship between medication adherence to psychotropic agents and psychiatric symptoms in Syrian refugees who were followed in a community health center in Turkey.

The aim of this study was written in a separate paragraph with details (lines 86-92).

Materials and Methods 

1- All subsections of methods should be numbered. Study design and participants can be described in a unique section.

All subtitles were numbered; the study design and participants were described differently.

2- Data analysis should be better renamed as "Statistical analysis."

The data analysis section was renamed "Statistical Analysis." (line 173).

Results and discussions 

The results section can be divided into several subsections. The first part, can be renamed as "Main characteristics of the total sample", with reference to Table 1.

We divided the results section into three parts “Main Characteristics of the total sample," "Correlation between GMAS-AR and BSI" and "Predictors of GMAS-AR" with references table 1-2-3, respectively.

Conclusion

1- Treatment adherence is low. I recommend to draw some recommendations about potential interventions.

We added some specific interventions and explained the next step with intervention in the conclusion part as recommendations. (lines 312-325).

Round 2

Reviewer 2 Report

The authors have substantially improved the paper. They have followed all the recommendations and responded adequately in the response letter.

I do not have any other questions or comments.